# Clinic-level complexities prevent effective engagement of people living with HIV who are out-of-care

Srija Dutta[1], Brendan H. Pulsifer[2], Kaylin V. Dance[3], Eric P. Leue[4], Melissa Beaupierre[4], Kennedi Lowman[5], Jessica M. Sales[6], Melanie Strahm[4], Jeri Sumitani[4], Jonathan A. Colasanti[3], Ameeta S. Kalokhe[1,3]*

1 Hubert Department of Global Health, Rollins School of Public Health, Emory University, Atlanta, GA, United States of America, 2 Emory School of Medicine, Atlanta, GA, United States of America, 3 Division of Infectious Diseases, Department of Medicine, Emory University School of Medicine, Atlanta, GA, United States of America, 4 Grady Health System, Atlanta, GA, United States of America, 5 THRIVESS, Inc., Atlanta, GA, United States of America, 6 Behavioral, Social and Health Education Sciences, Rollins School of Public Health, Emory University, Atlanta, GA, United States of America

* akalokh@emory.edu

**Data Availability Statement:** "Data cannot be made publicly available due to the potentially identifying nature of the interviews (i.e. some of the quotations from the qualitative interviews of the

## Abstract

Approximately half of people living with HIV (PLWH) in the United States are not retained in HIV care. Although numerous studies have identified individual-level barriers to care (i.e., substance abuse, mental health, housing, transportation challenges), less is known about institutional-level barriers. We aimed to identify clinic-level barriers to HIV care and strategies to address them to better engage PLWH who have been out of care (PLWH-OOC). As part of a larger qualitative study in a Ryan White-funded HIV Clinic in Atlanta, which aimed to understand the acceptance and feasibility of community-based HIV care models to better reach PLWH-OOC, we explored barriers and facilitators of HIV care engagement. From October 2022-March 2023, 18 in-depth-interviews were conducted with HIV-care providers, administrators, social workers, and members of a Community Advisory Board (CAB) comprised of PLWH-OOC. Transcripts were coded by trained team members using a consensus approach. Several clinic-level barriers emerged: 1) the large burden placed on patients to provide proof of eligibility to receive Ryan White Program services, 2) inflexibility of provider clinic schedules, 3) inadequate processes to identify patients at risk of disengaging from care, 4) poorly-resourced hospital-to-clinic transitions, 5) inadequate systems to address primary care needs outside of HIV care, and 6) HIV stigma among medical professionals. Strategies to address these barriers included: 1) colocation of HIV and non-HIV services, 2) community-based care options that do not require patients to navigate complex transportation systems, 3) hospital and community-based peer navigation services, 4) dedicated staffing to identify and support PLWH-OOC, and 5) enhanced systems support to help patients collect the high burden of documentation required to receive subsidized HIV care. Several systems-level HIV care barriers exist and intersect with individual and community-level barriers to disproportionately affect HIV care engagement among PLWH-OOC. Findings suggest several strategies that should be considered to reach the remaining 50% of PLWH who remain out-of-care.

participants may compromise their identities). Readers can contact Ameeta Kalokhe or the Emory Institutional Review Board (irb@emory.edu; 1-404-712-0720; https://www.irb.emory.edu/) to request the data. In line with PLOS ONE policy, readers who request the data will be provided with the de-identified dataset. The data will be stored long-term on an Emory secured shared drive.

**Funding:** This work was supported through an administrative supplement by the Center for AIDS Research at Emory University (P30AI050409; https://cfar.emory.edu/) to Ameeta Kalokhe and Jonathan Colasanti. The funders played no role in the study design, data collection, analysis, decision to pbulish nor preparation of the manuscript.

**Competing interests:** The authors have declared that no competing interests exist.

# Introduction

The Centers for Disease Control and Prevention (CDC) estimates that there are nearly 1.2 million people with HIV (PLWH) in the United States (US) [1], with the southern US being home to the highest rates of HIV infection and associated mortality in the country [2, 3]. Although only 38% of the US population resides in the South, HIV diagnoses in Southern states encompass over 50% of new cases every year [2]. The disproportionately high HIV-associated mortality and transmission rates have been attributed to lower rates of HIV care retention and viral suppression [4]. For example, in Georgia, the US state and study setting of the present study, only half (52%) of PLWH are retained in care and only 60% are virally suppressed [5], underscoring the importance of identifying and addressing barriers PLWH may face with adhering to care and antiretroviral therapy (ART).

Several studies have identified factors affecting retention of PLWH in HIV care in the South at various levels of the Socio-ecological Model (SEM) [6]. Commonly identified individual-level barriers include poor mental health, substance abuse, trauma history, and food and housing instability [7, 8]. Community- and societal/policy-level barriers in the South have also been well-studied and include: fragmented public transportation and poor geographic accessibility of HIV care, high community-level stigma around HIV and HIV risk behaviors, limited affordable housing availability, poor HIV and non-HIV health care infrastructure (including a shortage of HIV care providers), high levels of poverty, low levels of insurance, lack of Medicaid expansion, and HIV criminalization and prosecution laws [4, 9–11]. Institution-level factors, which include clinic and health care system structures, processes, and policies, also may influence retention in HIV care but have been far less studied [12, 13].

Ryan White-funded clinics (RWCs) provide HIV care to half the nation's population of PLWH, specifically those who are un/underinsured [14]. Importantly, many RWCs have onsite services that address many barriers to care through onsite case management, mental health, and substance abuse treatment services. Yet, the rates of disengagement from care suggest the need to examine potential clinic-level factors within RWCs that may affect patient retention in care.

As part of a larger mixed-methods study exploring the potential for community-based HIV care versus traditional clinic-based ("fixed clinic") care models to better reach, reengage and provide HIV care to PLWH who had been out-of-care, we examined multilevel factors impacting the success of these models. Through the qualitative exploration, participants identified several clinic-level barriers associated with fixed clinic care that were intersecting and additive to individual and community-level barriers. This manuscript describes these institutional barriers and potential strategies to address them.

# Methods

## Overview

This qualitative analysis is embedded within a larger mixed-methods research study that aimed to examine the potential acceptance, feasibility, and effectiveness of community-based mobile HIV care programs (i.e., home-based care, mobile clinic care) versus traditional fixed clinic care in re-engaging PLWH-OOC and barriers and facilitators influencing their implementation [15, 16]. This sub-study reports on the institution-level challenges that impeded effective HIV care provision to this subgroup. The study was approved by the Institutional Review Board of Emory University and the Grady Health System Research Oversight Committee. All participants provided written informed consent prior to participation. All study team members received training in research ethics.

## Participants

From November 2022 to March 2023, key stakeholders involved in mobile and fixed clinic HIV care programs were recruited to participate in in-depth interviews. Participants included: HIV care providers, clinic administrators, social workers, and members of a Community Advisory Board (CAB) established for the present study. The CAB was comprised of a diverse group of PLWH (with representation by racial, gender, and sexual minorities) who had experienced periods of being out-of-care at various points in their life and thus could speak to challenges with the mobile and fixed clinic models. Many brought experience of receiving HIV care in Atlanta, Georgia as well as other parts of the country over the course of their lifetimes. The CAB was established by two partner community-based partner organizations that aim to build collective HIV advocacy and support for Black same-gender-loving men and Black women. The CAB members and clinic providers, staff and administrators were recruited by email, phone call, and/or direct contact.

## Data collection

Interview guides were developed to explore factors impacting effective reach and care provision to PLWH-OOC via mobile clinics, home-based care, and traditional fixed HIV care. Two interview guides were created and administered to participants–one tailored to staff and providers, and the other tailored to CAB members. Pertinent to the present analysis, the guide included questions to investigate the perceived effectiveness, barriers, and facilitators of care coordination of HIV care, non-HIV subspecialty care, and support services. Participants were asked to compare and contrast the different models for HIV care delivery, and in doing so, commonly discussed clinic-level barriers to HIV care access, their intersections with individual- and community-level barriers, and strategies that could be employed to address them. In-depth interviews (IDIs) were conducted by a graduate research assistant, one-on-one, in a private, clinical setting or virtually via Zoom and audio-recorded. Interviews typically lasted between 30–60 minutes. Participants were compensated $50 for their participation.

## Data analysis

All IDIs were transcribed verbatim and checked for quality and to ensure they were void of identifying information. Trained members of the study team coded transcripts in parallel with conduct of the IDIs, allowing inductive codes to be further probed in subsequent interviews. The codebook was developed based on inductive and deductive codes after three initial interviews were conducted and refined over time to include new codes that emerged over the course of the study. Each interview was coded independently by two members of the study team using MAXQDA 2022 [17]. When discrepancies in coding arose, they were discussed with a third study team member until consensus was reached. For the present analysis, we identified themes pertinent to clinic-level factors impacting HIV care engagement among PLWH-OOC and strategies to address them and selected explanatory, representative quotes.

## Results

### Participant characteristics

A total of 18 participants were interviewed. Across participants, a diversity of clinical roles were represented: 7/18 (39%) were CAB members, 6/18 (33%) providers, and 5/18 (28%) were staff in administrative or social services. One-third (33% or 6/18) had been working in this role for over 7 years, 39% (7/18) for 4–6 years, and 28% (5/18) for 1–3 years. Additional role and demographic data was not collected to protect the identities of the participants.

## Barriers posed by systems complexities

Providers, staff, and CAB members all spoke to various elements of the healthcare system that impeded effective care provision to PLWH at risk of disengaging from care. Specific themes that arose include: 1) the large burden placed on patients to provide proof of eligibility to receive Ryan White Program services, 2) the inflexibility of provider clinic schedules not allowing for commonly encountered patient social challenges (i.e., transportation delays), 3) inadequate processes to identify patients at risk of disengaging from care, 4) hospital-to-clinic transitions being insufficiently resourced, 5) inadequate systems to address primary care needs outside of HIV care, and 6) HIV stigma persisting among medical professionals.

**Large burden placed on patients to provide proof of eligibility to receive Ryan White Program services.** Many participants spoke about the exhaustive burden of paperwork and coordination required to receive Ryan White care and support services being placed on patients. Remembering to complete annual paperwork required for receipt of subsidized HIV care and a lack of knowledge regarding paperwork requirements were largely cited as examples of how this burden manifests within patients.

> *"They haven't said it like this, but the executive functioning involved in making sure that you're signed up for the right programs, like that you're on, ADAP [AIDS Drug Assistance Program], that your Ryan White [paperwork] is up to date. You know, and then remembering to call in your prescription three to five days before it's due so that you can either come pick it up or get it mailed. Like these are things that take a lot for some patients to be able to do every month, for example, for the meds. And also some people just don't remember appointments, don't have a way to keep track of appointments, or are too disorganized."*–Administrative Staff

Patients are required to submit their Ryan White Program eligibility paperwork through their case managers to enroll in the program. One participant discussed the challenges imposed on patients because they often have to make two separate visits to see their HIV care provider and case manager.

> *"You can't expect and make the appointment for me to see the doctor and assume I'll just see the case management another day. . .many times referrals can get lost in the transition."*–CAB Member

Participants elaborated that challenges to presenting the large amount of paperwork included: 1) transportation barriers, 2) patients not being aware of all of the eligibility documents required or which of their documents were missing, 3) patients who were homeless wanting to minimize the things they carry (including documents) when traveling, and 4) the potential of self-reminders for HIV care appointments and Ryan White program eligibility requirements to compromise privacy.

> *"The patient doesn't come into the clinic because, you know, I mean they don't have transportation or they, you know, there's all this paperwork that needed to be done that is not, wasn't done and they didn't know they were supposed to do it."*–HIV Care Provider

> *"That's why a lot of people are not in care—because it's just about impossible. I mean because if I'm homeless you have to travel with what little bit is possible from point A to point B, and I for one ain't writing down in no journal about no appointment or nothing else that somebody else can read. Especially if I'm a very private person."*–CAB Member

**Inadequate processes to identify patients at risk of disengaging from care.** Some participants discussed the gap in dedicated processes to preemptively identify patients at risk of falling out of care and their risk factors for falling out of care to help inform efforts to address barriers to care to prevent care disengagement.

*"There is no structured process to assess as to why somebody fell through the gaps. There is no concerted effort, structure, or process. . . to identify some of the key indicators that would tell us that either somebody is about to fall out of care or has fallen out of care."*–HIV Care Provider

Participants also spoke about difficulties in being able to respond in a timely manner with support services when patients did demonstrate a gap in care or during times of crisis.

*"I think it happens quite often. . .situations where we need to offer immediate resources that we don't have or that we can't offer immediately and effectively to patients that need it in order to keep on their care regimen and adhere to their medication."*–HIV Care Provider

**Patient challenges with post-hospital follow-up.** In the context of PLWH who are out-of-care being hospitalized, participants spoke about the complexities of hospital-to-clinic follow-up. They discussed the challenges of establishing post-hospital follow-up, commonly citing difficulties with needing to access primary HIV care, other subspecialty care, plus social services.

*"I don't have time to worry about the nutritionist even though that's a big factor I need. . .referral processes and the many times referrals get lost in transition after hospital visits is something you can count as a barrier."*–CAB Member

Some providers discussed the lack of continuity of provider care during and post-hospitalization as a barrier to provision of effective HIV care.

*"So, [you as] the inpatient provider. . .caught them for the first time in six months. And as a provider with a full [outpatient] schedule, you just can't see everyone and at some point you– you would have to refer them to someone else or with [the home-based care program], but, like, then like who is the person that is following up with that? That would be their–their primary doctor. And they might not even know the patient was hospitalized right away."*–HIV Care Provider

**Inflexible clinic schedules and patient social barriers pose additive barriers to care.**
Lack of easily accessible appointment scheduling and timely appointments were repeatedly discussed as posing challenges to HIV care retention across all participant groups.

*"I think everybody has some level of a logistical barrier, whether it's financial or transportation or geographic. But underlying all of that in the vast majority of patients that fall, tend to fall out of care–like truly fall out of care, I think is–one is access to the clinic in terms like lack of ease. Like an easy way for them to get an appointment. Straightforward, you know, timely, whatever."*–Administrative Staff

Another participant further elaborated on how social challenges like transportation, limited appointment availability, and clinic schedule inflexibility were additive in posing barriers to care access.

*"We can start off with lacking access to transportation or reliable transportation. What I mean by that is even public transportation can be a challenge. Delays will cause the patient to be late for an in-person appointment, and then that appointment gets cancelled, and then the provider is not available and they have to try again, which then gets frustrating. The patient, then, ultimately gives up."*–Social Worker

Some explained that the process of just getting to-and-from the clinic often involved navigating complex transportation systems and delays from both patients and providers that result in patients having to sacrifice multiple hours of their day to complete their visit.

*"They check your blood, check your breathing, ask you questions, and you have to wait for your doctor and stand in a long pharmacy line for meds before you are out. And your whole day is gone if you're like me. I'm old and my whole day is done just getting there."* ¬ CAB Member

Participants across all roles commonly cited the instrumental role of stable housing when aiming to re-engage and retain an individual into care. Housing insecurity was reported as a barrier to both fixed clinic and community-based HIV care.

*"If they don't have a solid location such as housing or a home to go to, that could be another issue, 'cause there's no home for y'all to go to and provide care with home-based care. And, secondly, they're not gonna be trying to get to a mobile unit or clinic for their health. They're gonna be trying to get to someplace that's gonna help them with housing. So that could be another barrier."*–CAB Member

**Inadequate systems to address primary care needs outside of HIV care.** One provider raised concern with systems not being in place to address primary care needs outside of routine HIV care, such as efforts to ensure patient vaccines were up-to-date. This included a lack of reminders for patients to schedule subsequent appointments and for providers to order the vaccines.

*"And you know, Hep B and Hepatitis A vaccines have to be delivered [in a series]. . .at baseline and then you get one in one month and one in six months. I can't even tell you the number of times that patients never show up for subsequent appointments, because they just forget, or providers forget to order the vaccine or make them an appointment. . .you can't just expect patients to remember."*–HIV Care Provider

**Persistent HIV stigma among medical professionals.** A CAB member discussed how persistent HIV stigma and limited awareness of what it means to live with HIV today among medical professionals deterring patients from accessing care. The participant speculated this was due to educational gaps and influence of the political environment.

*"But again most–sometimes it's the people who are caring for you that comes with the stigma, that stigmatize you. It's not the people in the grocery store. A lot of times it's the people who are taking care of you. I've heard some of the most idiotic stuff come out of the mouths of the*

*medical professional with regard to my HIV care. So I'll throw this all the way back to our educational system, our educational/political system. Because HIV isn't going anywhere I personally don't believe not a single medical professional should graduate from a college without beyond basic knowledge of HIV and HIV care."*–CAB Member

### Proposed strategies to address systems complexities to HIV care

Participants discussed key strategies to address systems complexities and challenges that deter patients from seeking care: 1) co-location of HIV and non-HIV services, 2) community-based care options that do not require patients to navigate complex transportation systems, 3) peer navigation services to help patients maneuver and coordinate their care between clinics and support service organizations, 4) dedicated staffing to identify PLWH-OOC, and 5) enhanced systems support for patients to help them collect and submit the high burden of program documentation required to receive subsidized HIV care and support services.

**Colocation of HIV and non-HIV services.** Many participants discussed how colocation of services ("one-stop shops") could enhance retention in care by providing convenience, reducing the need to travel to different clinics at different times, and helping overcome complex systems barriers to care access.

*"Because I feel like it's so important to put it all together for people, especially with this population. They can't just go walking and bouncing around to all these different offices."*–Administrative Staff

*"I think the more that you can try to co-localize specific services, you'll find that people are, I think, willing–because it's just–it becomes a one stop shop, right? It's convenient."*–HIV Care Provider

One CAB member discussed how ideally HIV care (i.e., medical visit, laboratories) would be collocated with non-HIV services (i.e., mental health services) for ease.

*"In my perfect world, I would want–I know this is a far reach, but I would love to be able to do labs and then see the provider and it be completely up to date at that moment instead of having to do two trips to do labs and then come back. . . .I love that one-stop shop. So after I see the provider, then I can go on to mental health. . ."*—CAB member

**Community-based care helps circumvent challenges of poor public transportation systems and allows for greater visit flexibility.** Participants discussed community-based care would help patients sidestep transportation costs and time lost in maneuvering public transportation. One participant elaborated that such models could afford providers with greater flexibility in tailoring HIV care for vulnerable patients and help patients overcome challenges they face with the city's fragmented transportation system which requires them to transfer multiple train and bus lines to come to the HIV clinic.

*"I think that [the home-based HIV care program is] like highly necessary, especially just because we have such a poor public transportation system in Atlanta or in Georgia in general. And we have a lot of patients that live further out who have transportations difficulties and/or are not able to like take multiple MARTA buses or trains or whatever just to get here. And so I just think that this actually allows us the flexibility of particularly seeing people who are much more vulnerable."*–HIV Care Provider

Home-based care allowed patients to repurpose money they would have used for transportation to pay for other basic needs.

*"But even for our patients that are dealing with fixed incomes, some may have to pay for transportation if they came to the clinic. Well, [through home-based care], we're avoiding them spending money on going to spend on transportation, they can put it on their bill or food, because they're coming out to them."*–Administrative Staff

Home-based care also helped patients adhere to complex medical recommendations by having healthcare providers offer in-home instruction and clarification. It filled a gap in patient awareness of where and how to seek care and administer their medications.

*"Patients left to their own devices I feel are more confused about the way they should care for themselves when they're not in a hospital or a medical setting. When they're home and they have to sort their pills out themselves or inject themselves with their medication, they appreciate a medical individual or EMT or somebody coming to them showing them how to do these things, doing them for them, giving them that information that they just would not have gotten, or the clarity that they would have not gotten if they went and did it on a Google search engine, you know?"*–Administrative Staff

Lastly, one participant discussed how home-based care models allow for undivided provider attention and greater visit flexibility to effectively address patient needs by not being held to time limits often placed on clinic-based care.

*"In home I feel like you have more of a one-on-one undivided attention so you be able to have more open dialogue. You won't feel like it's all a time limit of being rushed no matter like if it is like 30 minutes. You don't feel that."*–CAB Member

**Importance of community health workers and peer navigators to manage systems complexities.** Many participants raised the need to use peer navigators and community health workers to overcome systems complexities in receiving care for people at risk of being disengaged. Participants discussed their role in outreach and care coordination, such as identifying and relinking those in the community who could not be contacted through other clinic means.

*"And peer navigators I think could also play a really big role. . . in looking at those patients, finding those patients that have been out of care."*–HIV Care Provider

*"Once they're kind of lost to care and if they're not connected to [the electronic medical record] we have to wait for them to show up. . .We do have a linkage navigator that goes to [the hospital], so if they happen to be admitted, luckily she can run a report and we know from that aspect. But if they don't, they're just lost to care, there's no way we can find them, especially if they don't have a next of kin where we can contact."*–Social Worker

*"That's why the linkage person being present is important. . .If they could have a company vehicle. . .I would link that person to the dentist right then and there and at least before I leave that home, try to have a scheduled appointment. And then if transportation was the issue, I come back and take you to your appointment."*–CAB Member

Another participant talked about how community health workers could help facilitate patient use of other services to meet basic needs and improve quality of life. The community

health workers could help assess needs and bring necessary services to patients, forgoing the need for patients to travel to multiple locations to receive care and support.

*"I think community health workers, certainly peer navigation, all of those things are useful, because again, same thing; like I think, you know, if a peer navigator or like a community health worker can go out and assess what other needs are necessary, be it like getting them enrolled for [nutrition/meal support Community Based Organization] if food insecurity is an issue, you know, having our nutritionist go out. And like just so many other, you know, have a physical therapist go out there if you need some help, I think are all sort of additional ways, again, to sort of improve the comprehensiveness of the care that we can give the patients so that they don't have to go to 15 different places to get their care."–HIV Care Provider*

A CAB member discussed the importance of the patient-peer relationship beginning at the patient's home along with the first visit. The peer could build patient comfort and sense of support and help the patient navigate to necessary services.

*"I would say the first steps would be–I would say start with a peer to make you feel like you have someone to be able to support you along the way. . . .to where they can like navigate these other services so it might be mental health services and things like that. So I feel like the introduction at home to be always with maybe that first appointment being with a peer or someone from the care team besides the doctor."–CAB Member*

**Dedicated staff to identify and reach out to people who have fallen out of care.** To help address deficiencies in methods of identifying patients who have fallen out of care, a CAB member suggested clinics should have dedicated personnel who recognize and respond when there are gaps in patient care visits.

*"Follow-up and outreach. Pretty much that. Now, if someone maybe is delegated to do that–I know maybe the medical staff may not be able to reach out to every patient or nurse practitioner may not be able to reach out to every patient, but there should be someone–maybe a liaison or someone in between–that could say, 'Hey. Oh, wow. Mister [name] hasn't been back. He's supposed to be back–okay.'"–CAB Member*

**Systems support to help patients submit necessary documentation for subsidized HIV care.** Participants suggested strategies to support patients in collection and submission of paperwork necessary to be eligible for HIV support services and subsidized care. They discussed utilizing members of the healthcare team to help inform patients about program eligibility documents they needed to complete and to collect and submit the documents on the patient's behalf.

*"One of the providers that I know well has like brought patients documents back to clinic so that they can enroll in, or do their financial counseling, or has gone out and said this is what you're going to need. But I think there has definitely been some like bringing documents back to help patients enroll in things, or [Nutrition Support Agency] forms, things like that. Which is good."–HIV Care Provider*

*"All of our patients, in order to come to the clinic have to submit Ryan White documents; they have to certify once a year. And because their [home-based care team] is going out to the*

*home, they collect some of those documents and bring them back to the clinic."*–Social
Worker

## Discussion

To increase the national HIV viral suppression rate among people diagnosed with HIV from
the present 66% [18] to the Ending the HIV Epidemic target of 95% by 2025 [19], a compre-
hensive, nuanced understanding of the barriers to longitudinal HIV care engagement is criti-
cal. While numerous studies have identified individual and community-level challenges with
HIV care retention [7, 8, 20, 21], few have examined clinical-level barriers and their interplay
with barriers at other levels of the SEM. To the best of our knowledge, this is the first study to
have examined clinic-level factors impacting retention in care from the perspective of
PLWH-OOC and other key clinic stakeholders simultaneously. In doing so, it reveals how the
multilevel factors intersect and potentiate the effects of one-another, increasing the overall
magnitude of care burden for PLWH-OOC, and also suggests strategies that could simulta-
neously impact multiple levels of the SEM (Fig 1).

Many of the clinic-level barriers noted in our study, such as limited appointment availabil-
ity [20–22], inflexible systems that do not allow for patient late arrivals [23], understaffed hos-
pital-to-clinic transitions [24–26], and negative patient-provider/staff interactions due to
stigma [7, 20, 21, 23] have been reported by others. Other studies have additionally identified
complicated automated scheduling systems [7] and healthcare fragmentation [20] (where not
all services like laboratories are provided during the care visit) as clinic-level barriers to care.
Our study adds to this literature by also uniquely identifying the patient burden of documenta-
tion and care coordination and gaps in clinic processes to preemptively identify patients at risk
of falling out of care (or who have fallen out of care) to enable early outreach as clinic-level bar-
riers. Individually, each of the clinic-level factors can serve as critical barriers to care, but when
layered atop individual- and societal-level barriers, the challenges with care engagement can
seem even more unsurmountable. For example, for an individual who lacks personal

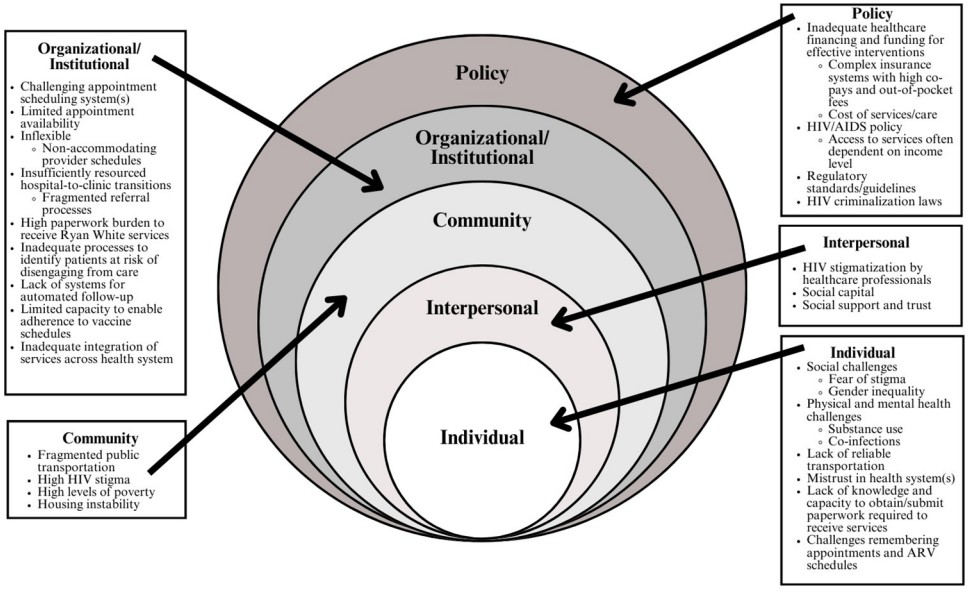

**Fig 1. Systems-level complexities illustrated using the socio-ecological model (SEM).**

transportation and must use public transportation, (which is often unreliable with multiple line transfers), a clinic with fully booked visits that is unable to accommodate late arrivals amplifies the retention barrier. For an individual with high levels of internalized HIV stigma or trauma, negative, judgmental interactions with clinic providers or staff may magnify anxiety and discomfort with care access. For an individual experiencing homelessness battling the competing priority of finding housing and disorganization associated with lack of permanent residency, the need to obtaining and present the significant amount of documentation necessary to receive Ryan White-subsidized HIV care can be overwhelming.

Solutions to overcome institutional barriers suggested by participants largely parallel those identified in other studies [27–29]. Themes from our study underscore the need to challenge the high-throughput, rigid scheduling of clinical templates now increasingly common to many health systems. This would allow scheduling flexibility to accommodate the needs of PLWH-OOC, including post-hospitalization and other acute care visits and late arrivals that may result from public transportation delays, patient work or child-care obligations (a barrier identified in other studies) [30], and/or psychosocial challenges (i.e., housing instability, mental health, and substance abuse disorders) that affect patient planning capacity necessary to reach HIV care visits. Additionally, open access models provide flexibility within clinical settings, where adhering to an appointment time is less important [29]. By offering services directly through mobile clinics and home-based care programs, these models allow patients to redirect and repurpose time and money they would have used for transportation for other basic needs. Similar sentiments have been echoed by patients given the option of accessing Long-Acting Injectable Therapies (LAIs) at home versus in clinic settings [31]. Findings from this study align with others [32] in suggesting the need for health systems to expand navigation assistance through peer navigation and community health worker services to counter the multitude of systems-level complexities and care coordination burden placed on patients, and for funding agencies to consider policy modifications to reduce the burden of eligibility paperwork necessary for receipt of subsidized services. Lastly, continued public health awareness of HIV today across the health system, (e.g., by peers and popular opinion leaders, incorporating patient testimonials as suggested by a recent systematic review [33]), is critical to combat stigma to ensure HIV care is a comfortable, safe space and experience for those anxious of returning to care.

Participants across role (e.g., patient, providers, staff) largely aligned in their understanding of care barriers and strategies to enhance patient retention in HIV care with a few exceptions. Patients elaborated in greater detail on how community- and clinic-level barriers intersected and spoke to pervasive provider HIV stigma as a barrier, while providers and staff elaborated on specific clinic-level operational barriers visible from their "in-house" perspective. For example, patients discussed the compounding effects of fragmented city transportation systems, attempts by patients experiencing homelessness to minimize the paperwork they had to lug around, and the high burden of paperwork necessary for patients to present to be eligible for Ryan White services; providers and staff, in contrast, expanded upon rigid clinic scheduling and gaps in outreach systems. These nuanced differences gave critical depth to the analysis and highlighted the importance of our having included participants across roles in the study design. While the aim of the study was to provide a comprehensive identification of care barriers faced by patients, future studies should explore the differences noted by participant role in greater detail to ensure solutions addressing the greatest barriers to care are prioritized by clinics.

A key study strength included exploration of clinic-level barriers from multiple clinic stakeholders, including PLWH who have presently or in the recent past been out of care. A study limitation was that the study question was embedded in a broader exploration of factors

influencing acceptance of community-based versus traditional fixed clinic HIV care models, thus the parent study was not specifically designed to focus on clinic-level barriers and strategies to overcome them. Nonetheless, in this context, clinic-level barriers to care and methods to overcome them were discussed by several participants across participant role. An additional limitation was that the study was conducted in one urban center in the Southeast, which may affect transferability of study findings to other settings. Nonetheless, the CAB members who participated in the study had received HIV care in various clinic settings over time, suggesting many of the reported challenges may be common across settings.

In conclusion, providers and users of the systems alike, identify a need for multidisciplinary care teams that is closer to the community with co-located and ideally integrated services, adaptable appointment and provider availability, easier paperwork requirements, with overarching program coordination that can proactively identify those likely to fall out of care to provide a priori support or quickly identify gaps in care and provide system-level support for re-engagement. Ryan-White and Ending the HIV Epidemic funding streams generally support these approaches but those must be reinforced and funded to levels where staffing ratios can mirror the studies that demonstrate the efficacy of many of these approaches. Stigma within the healthcare setting is intolerable and must be addressed through education that reaches beyond the infectious diseases and HIV specialties through all reaches of staff such that a PLWH never encounters misinformation nor perceived stigma during encounters with their healthcare system.

## Acknowledgments

We express utmost gratitude toward each of the individuals who participated in the study and openly shared their perspectives with us.

## Author Contributions

**Conceptualization:** Srija Dutta, Brendan H. Pulsifer, Kennedi Lowman, Jessica M. Sales, Melanie Strahm, Jonathan A. Colasanti, Ameeta S. Kalokhe.

**Data curation:** Srija Dutta, Brendan H. Pulsifer, Kaylin V. Dance, Eric P. Leue, Melissa Beaupierre, Kennedi Lowman, Jeri Sumitani, Jonathan A. Colasanti, Ameeta S. Kalokhe.

**Formal analysis:** Srija Dutta, Brendan H. Pulsifer, Jonathan A. Colasanti, Ameeta S. Kalokhe.

**Funding acquisition:** Jonathan A. Colasanti, Ameeta S. Kalokhe.

**Methodology:** Srija Dutta, Brendan H. Pulsifer, Kaylin V. Dance, Kennedi Lowman, Jessica M. Sales, Jonathan A. Colasanti, Ameeta S. Kalokhe.

**Project administration:** Srija Dutta, Brendan H. Pulsifer, Kaylin V. Dance, Eric P. Leue, Melissa Beaupierre, Kennedi Lowman, Melanie Strahm, Jeri Sumitani, Jonathan A. Colasanti, Ameeta S. Kalokhe.

**Writing – original draft:** Srija Dutta, Ameeta S. Kalokhe.

**Writing – review & editing:** Srija Dutta, Brendan H. Pulsifer, Kennedi Lowman, Jonathan A. Colasanti, Ameeta S. Kalokhe.

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
