## [Decision Letter · Decision Letter 0]

26 Mar 2024

PONE-D-24-03550Clinic-level complexities prevent effective engagement of people living with HIV who are out-of-carePLOS ONE

Dear Dr. Kalokhe,

Thank you for submitting your manuscript to PLOS ONE. After careful consideration, we feel that it has merit but does not fully meet PLOS ONE’s publication criteria as it currently stands. Therefore, we invite you to submit a revised version of the manuscript that addresses the points raised during the review process.

**ACADEMIC EDITOR's comments: **1. Introduction: The introduction should be revised to include some literature reviews on HIV retention in line with the study objectives. The review should identify what has been done by other studies, state any gaps that may exist and how your study seeks to bridge these gaps and contribute to the body of knowledge on the topic.2. Methodology: It was stated that the study design is a qualitative analysis embeded within a larger mix-method design. However, a review shows this study to be largely qualitative in nature, if there is data to support a mixed-method design (qualitative/quantitative), please provide it, especially related to coding of qualitative data.

We look forward to receiving your revised manuscript.

Kind regards,

Moses Katbi, MD, MPH, MBA, DrPH, FRSPH

Academic Editor

PLOS ONE

“This work was supported through an administrative supplement by the Center for AIDS Research at Emory University (P30AI050409). We express utmost gratitude toward each of the individuals who participated in the study and openly shared their perspectives with us.”

“This work was supported through an administrative supplement by the Center for AIDS Research at Emory University (P30AI050409; https://cfar.emory.edu/) to Ameeta Kalokhe and Jonathan Colasanti. The funders played no role in the study design, data collection, analysis, decision to pbulish nor preparation of the manuscript.”

3. In this instance it seems there may be acceptable restrictions in place that prevent the public sharing of your minimal data. However, in line with our goal of ensuring long-term data availability to all interested researchers, PLOS’ Data Policy states that authors cannot be the sole named individuals responsible for ensuring data access (http://journals.plos.org/plosone/s/data-availability#loc-acceptable-data-sharing-methods).

Reviewers' comments:

Reviewer's Responses to Questions

**Comments to the Author**

1. Is the manuscript technically sound, and do the data support the conclusions?

Reviewer #1: Yes

Reviewer #2: Yes

2. Has the statistical analysis been performed appropriately and rigorously? 

Reviewer #1: Yes

Reviewer #2: Yes

3. Have the authors made all data underlying the findings in their manuscript fully available?

Reviewer #1: Yes

Reviewer #2: No

4. Is the manuscript presented in an intelligible fashion and written in standard English?

Reviewer #1: Yes

Reviewer #2: Yes

5. Review Comments to the Author

Reviewer #1: 1. Summary of the research

This qualitative study explored barriers and facilitators to HIV care engagement in an urban setting in Atlanta, Georgia. Eighteen in-depth interviews were conducted with HIV service providers, administrators, social workers and people living with HIV who have experienced being out of HIV care. The authors reported a number of clinic level barriers, including high coordination burden on patients, inflexible clinic schedules, issues with identification of patients at risk of disengaging, poorly resourced hospital-to-clinic transitions, challenges with follow up, and stigma by health professionals. They authors identified strategies to address the barriers, including co-location of services, community-based care options, use of peer navigators, dedicated staff to support PLWH and support in collecting documentation for subsidized care.

The authors claim that this study adds to the body of evidence because it is the first study to examine clinic level factors that impact on retention in care, and it identifies burden of documentation and gaps in clinic processes.

The manuscript abstract, introduction and methods are well written. However, the findings will do with some editing to improve flow. Some of the results statements in the findings are too summarized and may need to be augmented to better reflect the participant discussions.

My overall recommendation is that the authors edit the findings section to improve flow, and augment some of the results statements.

2. Examples and evidence

2.1. Major issues

2.1.1. The results section does not flow well, with concepts being mixed under some sub-sections where they don’t seem to belong. For example, a section describing high care coordination (starting line 144) speaks to transport issues and prescription issues, which are covered under other sub-sections. Furthermore, the section generally does not flow well and has some language issues that need address. I would like to suggest that the whole section be edited to address the above issues and improve its flow.

2.1.2. A number of results statements seem to be too summarized and may not have adequate content. They may need to be augmented with more content to better reflect the discussions and what is provided in the quotations. The result statements that may need to be revised include those in the following: line 165-166; line 186-188; line 225; line 263-264; line 273-274; line 293-294; line 317-320; line 337; line 347-348; line 360; line 375-376; line 388-389; line 398-399; and line 409-410.

2.1.3. The results section participant characteristics (line 1329-132) do not include baseline characteristics. I would suggest that these be added.

2.2. Minor issues

2.2.1. Difference tenses are used in different parts of the manuscript, varying from present to past tense. It would be better to harmonize the tenses used throughout the manuscript.

2.2.2. The abstract is generally well written, but the following may need address:

• Since this is a study on engagement of out of care PLWH, it may be better for the first sentence (line 26-27) to be written such that it reflects the negative situation in the study, so that it refers to those who not retained in HIV care.

• The word “automated” in line 40 may be better dropped because it does not seem to be supported by the findings.

2.2.3. The introduction is well written. However, different viral suppression levels are quoted in the introduction (line 58, reference 5) compared to what is in the discussion (line 425). Consider harmonizing, otherwise if there is need to quote both, provide clear justify for that.

2.2.4. The research methods are well written and technically sound. The following minor issues may however need address:

• The sentence starting “In comparing and contrasting …… “ in line 110-113 sounds more like a funding than part of the methods. If it is to be retained, it may be better to rewrite it so that it sounds more of the methods.

• Though it is mentioned in the results section that the research team was trained, it may be better to also mention it in the methods that the research assistants were trained especially in research ethics.

• Please correct the typographical error one-one-one in line 114.

• In line 119, you may want to rephrase the part of the sentence reading “devoid of identifying information”.

2.2.5. The results section does not flow well. I would suggest that you consider revising as follows:

• For point 5 in line 139-141, it may be better to use the description as provided in line 262 but possibly without the word “automated” which does not seem to be supported by the findings.

• It is not clear why the phrase “care coordination” is used in line 145 as it does not seem to be supported by the presented findings.

• The bulk of what is in the quotation from line 151-158 does not seem relevant to the header (line 144). If it is to be kept, it may be better to ensure that it is reflected elsewhere. I would suggest you consider deleting “Transportation, co-pays” (line 151) and the rest of the quotation from “You know, and then ….” right up to the end of the quotation (line 153-158).

• The quotation in line 160-163 does not seem relevant to this section and may need to be moved to a sub-section that relates to referrals.

• The statement in line 165-166 in addition to speaking to program eligibility paperwork, also speak to transport and knowledge gap issues, the latter two issues which are not related to care coordination issues that the sub-section header refers to (line 144). The quotation also speaks to both transportation and paperwork issues. For better flow of this sub-section, I would like to suggest that only issues related to this sub-heading (line 144) be included while other issues and quotations would be better moved to relevant sub-sections.

• The results statement in line 178-179 is not very clear and the author could consider revising it.

• There does not seem to be adequate evidence provided to support the first half of the statement in line 195-196 as the quotation 199-202 does not adequately refer to it. It could be helpful to provide another quotation to provide supporting evidence for that first part of the statement.

• The second part of the sentence that starts “…. Their feedback highlighting now …..” in line 253-254 seems unclear. The authors could consider revising or deleting it.

• The sub-heading in line 262 includes the word “automated”, but there does not seem to be findings to support its inclusion. I would suggest that the authors consider dropping it if there are no findings to support its inclusion. Furthermore, line 262 is very broad while the quotation provided is very specific to issues around hepatitis vaccination. I would suggest adding findings or quotations to support the statement.

• The results statement in line 304-306 and the quotation in line 308-315 do not seem to be well aligned, and the conclusion may therefore be an overreach of the findings. I would suggest that the authors consider using quotations that more clearly talk to co-location, and not just the need for the other non-HIV services referred to.

• There is need to consider replacing the word “longitudinal” in the statement in line 388-389 to possibly make this statement clearer.

2.2.6. The discussion section is generally well written though it could do with some editing to improve flow. A few issues that may need address include:

• In line 463, child-care obligations are mentioned, but do not seem to have been included in the study findings. The authors may want to clearly indicate that it’s a suggestion to include them though they were not study findings.

• The analogy “Long-Acting Injectable Therapies (LAIs) in advancing HIV care” in line 470 may not be the best for this situation and the authors could consider using a different analogy.

• The statement that starts “Nonetheless, the CAB members ….. “ in line 486-488 refers to the experience of CAB members in other clinic settings, but I am not sure that the finding was presented anywhere else in the manuscript. I would suggest as a finding it be included in the results section before it is referred to in the discussion section.

• The statement that stigma within health care settings must be addressed through education (line 496-497) may not be adequate. I would like to suggest that the authors consider including other strategies to me more complete.

Reviewer #2: The study was well conducted important details included. However, methodology did not include the demographics of the participants, I will be good to have that information.

Under the result, The study provides a unique opportunity to compare the perceived reason for dropping out of care from patients' and healthcare workers. But this was not well highlighted. It be good to compare and contrast this findings to see areas of agreement and areas where they differ This will help in understanding the deeper dynamics involved in deciding to drop out of care. An effort was made to do this but it fails to address it well. Distinct themes and sub themes for the CAB member and care providers'

The discussion should compare that findings with more similar studies and not just repeat what has already been written in the result.

6. PLOS authors have the option to publish the peer review history of their article (what does this mean?). If published, this will include your full peer review and any attached files.

Reviewer #1: **Yes: **Brian C Chirombo, MBChB, MPH

Reviewer #2: No

---

## [Author Response · Author response to Decision Letter 0]

29 Apr 2024

Response to Reviewers (also attached as Word Document)

Reviewer #1: 1. Summary of the research

This qualitative study explored barriers and facilitators to HIV care engagement in an urban setting in Atlanta, Georgia. Eighteen in-depth interviews were conducted with HIV service providers, administrators, social workers and people living with HIV who have experienced being out of HIV care. The authors reported a number of clinic level barriers, including high coordination burden on patients, inflexible clinic schedules, issues with identification of patients at risk of disengaging, poorly resourced hospital-to-clinic transitions, challenges with follow up, and stigma by health professionals. They authors identified strategies to address the barriers, including co-location of services, community-based care options, use of peer navigators, dedicated staff to support PLWH and support in collecting documentation for subsidized care.

The authors claim that this study adds to the body of evidence because it is the first study to examine clinic level factors that impact on retention in care, and it identifies burden of documentation and gaps in clinic processes.

The manuscript abstract, introduction and methods are well written. However, the findings will do with some editing to improve flow. Some of the results statements in the findings are too summarized and may need to be augmented to better reflect the participant discussions.

My overall recommendation is that the authors edit the findings section to improve flow, and augment some of the results statements.

Thank you for this feedback. The manuscript is now revised to reflect the specific changes requested below to improve flow and augment the Results.

2. Examples and evidence

2.1. Major issues

2.1.1. The results section does not flow well, with concepts being mixed under some sub-sections where they don’t seem to belong. For example, a section describing high care coordination (starting line 144) speaks to transport issues and prescription issues, which are covered under other sub-sections. Furthermore, the section generally does not flow well and has some language issues that need address. I would like to suggest that the whole section be edited to address the above issues and improve its flow.

Thank you for this comment and for giving us specific feedback on the Results statements and quotes that needed to be revised. We have significantly amended the entire Results section as suggested for clarity and flow.

2.1.2. A number of results statements seem to be too summarized and may not have adequate content. They may need to be augmented with more content to better reflect the discussions and what is provided in the quotations. The result statements that may need to be revised include those in the following: line 165-166; line 186-188; line 225; line 263-264; line 273-274; line 293-294; line 317-320; line 337; line 347-348; line 360; line 375-376; line 388-389; line 398-399; and line 409-410.

We had originally included abridged statements for succinctness, but have now expanded the text as depicted in the table below. We were cautious to not overinterpret what was stated by the participant.

Lines in original submission Revised lines in current submission

165-166 175-180

186-188 193-195

225 231-232

263-264 270-273

273-274 282-285

293-294 304-306

317-320 326-330

337 347-349

347-348 359-361

360 372-373

375-376 389-392

388-389 404-406

398-399 416-417

409-410 426-430

2.1.3. The results section participant characteristics (line 1329-132) do not include baseline characteristics. I would suggest that these be added.

Given the small study setting and potential for quotes to be identifying, we did not collect demographic data to protect the identities of the participants. We have now elaborated on the role data we did collect (in aggregate) and added this statement in response to the reviewer’s suggestion: “One-third (33% or 6/18) had been working in this role for over 7 years, 39% (7/18) for 4-6 years, and 28% (5/18) for 1-3 years. Additional role and demographic data was not collected to protect the identities of the participants.” (Lines 136-138).

2.2. Minor issues

2.2.1. Difference tenses are used in different parts of the manuscript, varying from present to past tense. It would be better to harmonize the tenses used throughout the manuscript.

Thank you for noting this. We reviewed the manuscript and revised sentences (i.e., lines 90, 369) to align with the past tense used elsewhere. The Discussion, which discussions implications of the study and draws conclusions for the future, is left in present and future tense.

2.2.2. The abstract is generally well written, but the following may need address:

• Since this is a study on engagement of out of care PLWH, it may be better for the first sentence (line 26-27) to be written such that it reflects the negative situation in the study, so that it refers to those who not retained in HIV care.

• The word “automated” in line 40 may be better dropped because it does not seem to be supported by the findings.

The abstract has been revised to state “Approximately half of people living with HIV (PLWH) in the United States are not retained in HIV care” (line 25-26). “Automated” has been removed. The statement now reads “inadequate systems for primary and HIV care follow-up”(lines 39-40).

2.2.3. The introduction is well written. However, different viral suppression levels are quoted in the introduction (line 58, reference 5) compared to what is in the discussion (line 425). Consider harmonizing, otherwise if there is need to quote both, provide clear justify for that.

The Introduction cites the viral suppression level of 60% in Georgia (a state located at the epicenter of the US HIV epidemic in the southern region of the country). The discussion cites the national viral suppression level of 66% and is an aggregate figure of all states in the country. To help clarify this for the global PLOS ONE readership, we have added a line in the Introduction that describes and gives more context to Georgia (lines 57-58).

2.2.4. The research methods are well written and technically sound. The following minor issues may however need address:

• The sentence starting “In comparing and contrasting …… “ in line 110-113 sounds more like a funding than part of the methods. If it is to be retained, it may be better to rewrite it so that it sounds more of the methods.

This is now revised to read, “Participants were asked to compare and contrast the different models for HIV care delivery, and in doing so, commonly discussed clinic-level barriers to HIV care access, their intersections with individual- and community-level barriers, and strategies that could be employed to address them.” (Lines 114-115)

.

• Though it is mentioned in the results section that the research team was trained, it may be better to also mention it in the methods that the research assistants were trained especially in research ethics.

This has now been added (lines 93-94).

• Please correct the typographical error one-one-one in line 114.

Thank you. This has been corrected to read “one-on-one” (line 118).

• In line 119, you may want to rephrase the part of the sentence reading “devoid of identifying information”.

Thank you. The term “devoid” has been replaced with “void.” (Line 122).

2.2.5. The results section does not flow well. I would suggest that you consider revising as follows:

• For point 5 in line 139-141, it may be better to use the description as provided in line 262 but possibly without the word “automated” which does not seem to be supported by the findings.

These two lines are now congruent and read, “Inadequate systems to address primary care needs outside of HIV care” (lines 146 and 269)

• It is not clear why the phrase “care coordination” is used in line 145 as it does not seem to be supported by the presented findings.

This has been revised to read “coordination” (line 151)

• The bulk of what is in the quotation from line 151-158 does not seem relevant to the header (line 144). If it is to be kept, it may be better to ensure that it is reflected elsewhere. I would suggest you consider deleting “Transportation, co-pays” (line 151) and the rest of the quotation from “You know, and then ….” right up to the end of the quotation (line 153-158).

The section header has been edited to better reflect the content (lines 149-150). We deleted the “transportation, copays,” but retained the remainder of the quote because it explains the tremendous burden put on patients to receive subsidized care (i.e., paperwork for ADAP, paperwork for RW, remembering to call in prescriptions ahead of time, remembering and tracking appointments (as HIV care visits are also necessary for Ryan White eligibility).

• The quotation in line 160-163 does not seem relevant to this section and may need to be moved to a sub-section that relates to referrals.

More context to the quote has been provided to explain its relevance to this section (lines 166-169). The quote has also been trimmed to retain content most relevant.

• The statement in line 165-166 in addition to speaking to program eligibility paperwork, also speak to transport and knowledge gap issues, the latter two issues which are not related to care coordination issues that the sub-section header refers to (line 144). The quotation also speaks to both transportation and paperwork issues. For better flow of this sub-section, I would like to suggest that only issues related to this sub-heading (line 144) be included while other issues and quotations would be better moved to relevant sub-sections.

In the sentences preceding the quote (lines 175-180) we have now added more context to explain the relevance to the subject heading (which also was revised, lines 149-150).

• The results statement in line 178-179 is not very clear and the author could consider revising it.

The lines and subsequent quote have been removed as they are not directly relevant to the section header/theme.

• There does not seem to be adequate evidence provided to support the first half of the statement in line 195-196 as the quotation 199-202 does not adequately refer to it. It could be helpful to provide another quotation to provide supporting evidence for that first part of the statement.

The text has been revised to more accurately reflect the quote (lines 202-203).

• The second part of the sentence that starts “…. Their feedback highlighting now …..” in line 253-254 seems unclear. The authors could consider revising or deleting it.

This is now revised for clarity (lines 260-261).

• The sub-heading in line 262 includes the word “automated”, but there does not seem to be findings to support its inclusion. I would suggest that the authors consider dropping it if there are no findings to support its inclusion. Furthermore, line 262 is very broad while the quotation provided is very specific to issues around hepatitis vaccination. I would suggest adding findings or quotations to support the statement.

This has been revised (line 269).

• The results statement in line 304-306 and the quotation in line 308-315 do not seem to be well aligned, and the conclusion may therefore be an overreach of the findings. I would suggest that the authors consider using quotations that more clearly talk to co-location, and not just the need for the other non-HIV services referred to.

The quote and preceding text have been replaced with another quote discussing preference for colocation of services (lines 319-324).

• There is need to consider replacing the word “longitudinal” in the statement in line 388-389 to possibly make this statement clearer.

This is now deleted for clarity.

2.2.6. The discussion section is generally well written though it could do with some editing to improve flow. A few issues that may need address include:

• In line 463, child-care obligations are mentioned, but do not seem to have been included in the study findings. The authors may want to clearly indicate that it’s a suggestion to include them though they were not study findings.

This statement has now been revised and is supported with a citation (Lines 483-484).

• The analogy “Long-Acting Injectable Therapies (LAIs) in advancing HIV care” in line 470 may not be the best for this situation and the authors could consider using a different analogy.

This has been revised to better reflect why it is a suitable analogy (lines 488-491).

• The statement that starts “Nonetheless, the CAB members ….. “ in line 486-488 refers to the experience of CAB members in other clinic settings, but I am not sure that the finding was presented anywhere else in the manuscript. I would suggest as a finding it be included in the results section before it is referred to in the discussion section.

We have added to the Methods more description about the CAB members and where they received HIV care (lines 102-104).

• The statement that stigma within health care settings must be addressed through education (line 496-497) may not be adequate. I would like to suggest that the authors consider including other strategies to me more complete.

Additional details and a citation have been added (lines 496-497).

Reviewer #2: 

The study was well conducted important details included. However, methodology did not include the demographics of the participants, I will be good to have that information.

Given the small study setting and potential for quotes to be identifying, we did not collect demographic data to protect the identities of the participants. We have now elaborated on the role data we did collect (in aggregate) and added this statement in response to the reviewer’s suggestion: “One-third (33% or 6/18) had been working in this role for over 7 years, 39% (7/18) for 4-6 years, and 28% (5/18) for 1-3 years. Additional role and demographic data was not collected to protect the identities of the participants.” (Lines 136-138).

Under the result, The study provides a unique opportunity to compare the perceived reason for dropping out of care from patients' and healthcare workers. But this was not well highlighted. It be good to compare and contrast this findings to see areas of agreement and areas where they differ This will help in understanding the deeper dynamics involved in deciding to drop out of care. An effort was made to do this but it fails to address it well. Distinct themes and sub themes for the CAB member and care providers'

Thank you for this insightful comment. In response, we rereviewed the perspectives across participant role and have added the following text to the Discussion (lines 500-513). 

The discussion should compare that findings with more similar studies and not just repeat what has already been written in the result.

We have now expanded the number of similar studies discussed in the Discussion (483-499).

---

## [Editor Report · Decision Letter 1]

14 May 2024

Clinic-level complexities prevent effective engagement of people living with HIV who are out-of-care

PONE-D-24-03550R1

Dear Kalokhe Ameeta,

We’re pleased to inform you that your manuscript has been judged scientifically suitable for publication and will be formally accepted for publication once it meets all outstanding technical requirements.

Kind regards,

Moses Katbi, MD, MPH, MBA, FRSPH

Academic Editor

PLOS ONE
---

## [Editor Report · Acceptance letter]

21 May 2024

PONE-D-24-03550R1 

PLOS ONE

Dear Dr. Kalokhe, 

I'm pleased to inform you that your manuscript has been deemed suitable for publication in PLOS ONE. Congratulations! Your manuscript is now being handed over to our production team.

Kind regards, 

on behalf of

Dr. Moses Katbi 

Academic Editor

PLOS ONE